# LANGUAGE GANS FALLING SHORT

**Massimo Caccia**[*]
Mila, Université de Montréal
massimo.p.caccia@gmail.com

**Lucas Caccia**[*]
Mila, McGill University
lucas.page-caccia@mail.mcgill.ca

**William Fedus**
Mila, Université de Montréal
Google Brain, Montréal

**Hugo Larochelle**
Google Brain, Montréal
Mila, Université de Montréal
Canada CIFAR AI Chair

**Joëlle Pineau**
Mila, McGill University
Facebook AI Research, Montréal
Canada CIFAR AI Chair

**Laurent Charlin**
Mila, HEC Montréal
Canada CIFAR AI Chair

## ABSTRACT

Traditional natural language generation (NLG) models are trained using maximum likelihood estimation (MLE) which differs from the sample generation inference procedure. During training the ground truth tokens are passed to the model, however, during inference, the model instead reads its previously generated samples - a phenomenon coined *exposure bias*. Exposure bias was hypothesized to be a root cause of poor sample quality and thus many generative adversarial networks (GANs) were proposed as a remedy since they have identical training and inference. However, many of the ensuing GAN variants validated sample quality improvements but ignored loss of sample *diversity*. This work reiterates the fallacy of quality-only metrics and clearly demonstrate that the well-established technique of reducing softmax temperature can outperform GANs on a quality-only metric. Further, we establish a definitive *quality-diversity evaluation* procedure using temperature tuning over local and global sample metrics. Under this, we find that MLE models consistently outperform the proposed GAN variants over the whole quality-diversity space. Specifically, we find that 1) exposure bias appears to be less of an issue than the complications arising from non-differentiable, sequential GAN training; 2) MLE trained models provide a better quality/diversity trade-off compared to their GAN counterparts, all while being easier to train, easier to cross-validate, and less computationally expensive.[1]

## 1 INTRODUCTION

Generating fluent natural language is a central aim in Natural Language Processing (NLP). Transformer architectures (Vaswani et al., 2017) with hundreds of millions or billions of parameters regularly reestablish state-of-the-art on held-out validation perplexities, however, the generated samples can still often *lack coherence*. It was hypothesized that *exposure bias*, differences in the training and inference procedures, is the root cause for poor sample quality. Therefore, numerous GANs (Goodfellow et al., 2014) for text generation were proposed since they do not suffer from exposure bias due a training objective that directly seeks to improve sample quality (to a learned critic).

While many of the early GAN variants demonstrated improvements in sample quality (Yu et al., 2017; Guo et al., 2017), they often ignored loss of sample diversity. This is not only a theoretical concern because it is well-documented that GANs exhibit *mode collapse* (Che et al., 2016; Salimans et al., 2016) where the generator produces a subset of the "modes" in the training data, reducing

---

[*]Authors contributed equally

[1]Code to reproduce experiments is available at github.com/pclucas14/GansFallingShort

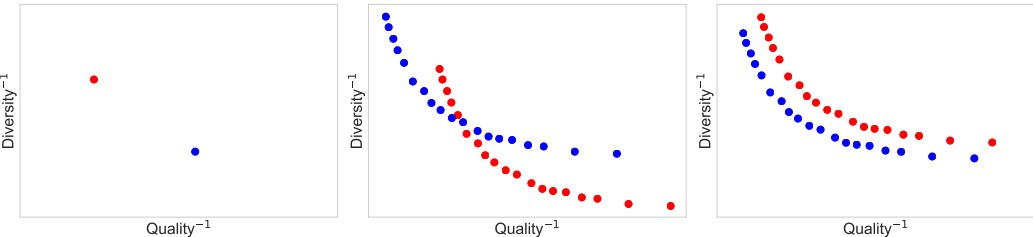

Figure 1: The importance of temperature for evaluating NLG models on quality *and* diversity. Each sub-figure plots inverse quality against inverse diversity (lower is better for both metrics). **Left:** current way of comparing NLG models. In this case, it is impossible to come to any meaningful conclusions about which model (red or blue) dominates the other. **Middle:** With our proposed NLG evaluation framework, the temperature sweep shines a light on the relative performance of the models: the red model should be used for high-diversity samples and the blue model for high-quality samples. **Right:** A second simulated scenario (consistent with the left Figure) where the temperature sweep reveals that the blue model dominates the red. That is, for any desired diversity-level, there is a temperature for which the blue model outperforms the red in terms of quality (and vice versa).

diversity. Evaluating generated samples remains challenging (Liu et al., 2016; Zhao et al., 2017), but Cífka et al. (2018) recommends assessing generated samples along two dimensions: 1) the *quality* of each sentence; 2) the *diversity* across sentences. However, this too is problematic because now the assessment relies on *two measures* which can make it difficult to compare models. Without further information about the relative trade-off between the two dimensions it is impossible to claim which is superior (Figure 1, left subplot).

In this work, we build on the natural relationship between a model's *softmax temperature* and the resultant *quality-diversity* trade-off: lower temperatures generate less diverse, higher-quality samples; higher temperatures increase the entropy of the distribution and produce more diverse, lower-quality samples. We propose a *temperature sweep* as a computationally efficient method to characterize the quality-diversity trade-off of different models. By explicitly controlling the temperature we remove a potential source of bias (models may have different "implicit" temperatures) and obtain a more complete understanding of each model's behavior (Figure 1 middle and right).

Additionally, we address that temperature modulation changes the entropy of the conditional distributions, as opposed to changing the entropy of the joint distribution. We explore other ways to navigate the quality-diversity space with less bias, including *stochastic beam search* and *generator rejection*. Although these methods provide small gains in the quality-diversity trade-off they have other computational limitations.

Despite the dizzying array of text-GAN variants and algorithms (Yu et al., 2017; Che et al., 2017; Lin et al., 2017; Zhang et al., 2017; Guo et al., 2017; Fedus et al., 2018; Lu et al., 2018a; Shi et al., 2018; Xu et al., 2018; Chen et al., 2018; Nie et al., 2019; d'Autume et al., 2019) our conclusions using temperature sweeps are clear: MLE models still dominate GANs. According to all metrics we studied, changing the temperature of MLE models at generation time leads to a better quality-diversity trade-off compared to GANs. This evaluation technique provides a definitive boundary for future GAN research and our hope is that it will help the NLP community accurately assess natural-language generation progress.

## 2 ADVERSARIAL TEXT GENERATION

GANs are implicit generative models learned via a competition between a generator network $G_\theta$ and a discriminator network $D_\phi$. The generator network $G_\theta$ produces samples from a probability distribution $p_{model}(x)$. The discriminator $D_\phi(x)$ attempts to distinguish whether an input value $x$ is real (training data) or generated. Mathematically, the GAN objective can be formulated as a minimax game

$$\mathcal{L} = \min_\theta \max_\phi \mathbb{E}_{x \sim p_{data}}[\log D_\phi(x)] + \mathbb{E}_{x \sim G_\theta}[1 - \log D_\phi(x)] \tag{1}$$

GANs were first proposed for continuous domains since their training procedure differentiates through the discriminator into the generator. However, modeling text, which is both discrete and typically modeled sequentially requires a challenging adaptation for GANs which, in their original formulation, were built upon "one-shot" continuous data. Discrete (sequential) data require an alternative approach. Yu et al. (2017) estimate the gradient to the generator via REINFORCE policy gradients (Williams, 1992). In their formulation, the discriminator evaluates full sequences. Therefore, to provide error attribution earlier for incomplete sequences and to reduce the variance of gradients, they perform $k$ Monte-Carlo rollouts until the sentence is completed.

Yu et al. (2017) advertise their model using two tasks which we argue (with hindsight) are flawed. First, they introduce a synthetic evaluation procedure where the underlying data distribution $P$ is known and can be queried. By representing $P$ with an LSTM (referred to as an oracle in the literature), they directly compute the likelihood of samples drawn from a generative model $G_\theta$. The issue is that they benchmark models against each other on this likelihood alone, i.e., the diagnostic is entirely blind to diversity. For example, a model that always outputs the same highly likely sequence would easily outperform other potentially superior models. For real data, there was no agreed-upon metric to evaluate the quality of unconditional NLG at the time. This led the authors to propose a new metric, *Corpus-level BLEU*, which computes the fraction of n-grams in a sample that appears in a reference corpus. Again, this metric is agnostic to diversity. Generating a single good sentence over and over will give a perfect BLEU score.

This paper sparked a stream of works that adopt their evaluation framework. Notably, RankGAN (Lin et al., 2017), MaliGAN (Che et al., 2017), TextGAN (Zhang et al., 2017), LeakGAN (Guo et al., 2017) and IRL-GAN (Shi et al., 2018) were proposed soon after.

All but one of the aforementioned papers evaluate sample quality alone. As a remedy, Zhu et al. (2018) propose a metric that compares a generated sentence with a corpus of generated sentences, called Self-BLEU. They, along with Lu et al. (2018b), provide an extensive comparison of GANs using quality (negative BLEU) and diversity (Self-Bleu). However, it is not clear which algorithm is superior, as evidenced by Figure 2, because no model simultaneously outperforms the other on both metrics. It is now standard for language GANs to evaluate simultaneously quality and diversity.

**GANs Trained without RL** Reinforcement Learning (RL) is often difficult to optimize, unstable, and sensitive to hyperparameters. Because of this, GAN-variants have recently been proposed that eschew RL-techniques in favor of fully-differentiable objectives. Cooperative training (CoT) (Lu et al., 2018a) and feature mover GAN (FM-GAN) Chen et al. (2018) are notable approaches.

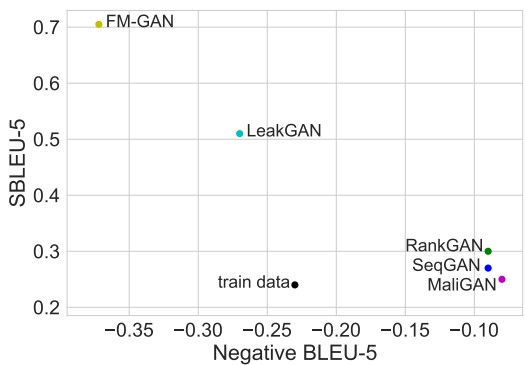

Figure 2: Negative BLEU-5 versus SBLEU-5 *(lower is better for both metrics)* on the EMNLP2017 News dataset taken from (Lu et al., 2018b) and this work (train data and FM-GAN). These scatter plots do not clearly show which algorithm is preferred since none strictly dominates on both metrics simultaneously.

Research in both RL and non-RL GANs is still very active (e.g., Xu et al. (2018); Nie et al. (2019); d'Autume et al. (2019); Li et al. (2019); Gagnon-Marchand et al. (2019)) and so our review is not exhaustive.

## 3 TEMPERATURE SWEEP: TOWARDS ROBUST NLG EVALUATION

During natural language generation, a single spurious sampled token can lead to an overall low-quality and incoherent sequence. In other words, a high-entropy conditional distribution may result in poor sample quality at inference. To address this problem, one can modulate the entropy of $G_\theta(x_t \mid x_{1:t-1})$ with a Boltzmann temperature parameter $\alpha$ (Ackley et al., 1988). If $o_t$ is the gen-

erator's pre-logit activation and $W$ is the word embedding matrix, then the conditional distribution of the generator is given by $G_\theta(x_t \mid x_{1:t-1}) = \text{softmax}(o_t \cdot W/\alpha)$. Decreasing $\alpha$ below 1.0 will increase $o_t$ and thus decrease the entropy of $G$'s conditional probability. Temperature tuning *naturally* moves the model in quality/diversity space. We demonstrate this in Table 1.

| $\alpha$ | Samples |
|---|---|
| 2.0 | (1) If you go at watch crucial characters putting awareness in Washington , forget there are now unique developments organized personally then why charge . |
| | (2) Front wants zero house blood number places than above spin 5 provide school projects which youth particularly teenager temporary dollars plenty of investors enjoy headed Japan about if federal assets own , at 41 . |
| 1.0 | (1) Researchers are expected to comment on where a scheme is sold , but it is no longer this big name at this point . |
| | (2) We know you ' re going to build the kind of home you ' re going to be expecting it can give us a better understanding of what ground test we ' re on this year , he explained . |
| 0.7 | (1) The other witnesses are believed to have been injured , the police said in a statement , adding that there was no immediate threat to any other witnesses . |
| | (2) The company ' s net income fell to 5 . 29 billion , or 2 cents per share , on the same period last year . |
| 0.0 | (1) The company ' s shares rose 1 . 5 percent to 1 . 81 percent , the highest since the end of the year . |
| | (2) The company ' s shares rose 1 . 5 percent to 1 . 81 percent , the highest since the end of the year . |

Table 1: The effect of temperature on samples from an language model trained via MLE on the EMNLP17 News dataset. At a temperature of $\alpha = 1.0$ the samples are syntactically correct but often lack in global coherence. The sample quality varies predictably with temperature. At $\alpha > 1.0$, the syntax breaks down and at $\alpha = 0.0$ the model always outputs the same sequence. At $\alpha = 0.7$ the samples are both of high quality and of sufficient diversity.

The evaluation protocol for NLG explained in Section 2 is to compare models with respect to both quality and diversity metrics. Often this results in a situation where it is impossible to tell which model is superior, as shown in Figure 2 and further exemplified in Figure 1 (Left). One can then control the quality-diversity trade-off of autoregressive text generators using temperature. We can leverage this tool to design a new evaluation framework that shines a light on the real performance of each model. More precisely, we propose to generate samples at s temperatures for each model in order to compute temperature curves in quality-diversity space. This is exemplified in Figure 1. We refer to this procedure as *temperature sweep*. This new way of evaluating NLG models allows practitioners to answer questions such as: which model to use if interested in high quality (or diversity) samples? Does a new model improve upon others in the quality/diversity space or is it just reducing the entropy of the distribution? It could also be leveraged as a cross-validation tool e.g., early-stop once the best temperature curve is achieved according to a heuristic.

In the next section we show, using temperature sweep, that MLE models consistently outperform the new proposed GAN variants everywhere in the quality-diversity space. MLE performs equally on synthetic data to CoT and outperforms it on real data, whilst being computationally and algorithmically less complicated. Our results are further validated in independent follow-up works (d'Autume et al., 2019; Alihosseini et al., 2019)where temperature sweeps show MLE outperforming GANs.

To implement temperature sweep we have taken advantage of the fact that our model factorizes the joint distribution over an observation as a product of conditional distributions over single tokens given all previous tokens (this is a property of most autoregressive neural networks). We change the temperature of these conditionals, which is straightforward to implement. However, this is different from changing the temperature of the joint probability distribution. The samples generated in this way are biased towards having lower entropy at the beginning of sentences compared to samples obtained by changing the entropy of the joint distribution. Changing the entropy of the joint distribution quickly becomes intractable with respect to vocabulary size and/or the total number of timesteps. We show below some entropy reducing techniques are less biased than changing the temperature of the conditionals and might achieve a better quality-diversity trade-off. However, we argue that theses methods aren't suitable for our evaluation protocol due to computational inefficiency (see Section 5.4).

**Stochastic Beam Search** is a popular approximate decoding technique. It is computationally expensive but leads to higher-likelihood samples than greedy decoding. Here, we focus on its *stochastic version* (not to be confused with the locally optimal one *local beam search*). In Stochastic beam search with beam size $k$, the $k$ most likely hypotheses (sampled so far) are kept at every decoding steps. In Appendix A, we detail this technique and explain how it is less biased than temperature tuning regarding the reduction of the joint distribution's entropy. Finally, the more traditional local (deterministic) beam search removes all sample diversity and is not useful in our setting.

**Generator Rejection Sampling** works as follow: generate some sentences; compute their likelihood under the generator's own distribution; accept/reject the sample given a threshold. The threshold enables the practitioner to modulate the quality-diversity trade-off. In Appendix B, we explain how it is less biased than temperature tuning, why it is computationally expensive, and how the discriminator can be used in rejection sampling.

## 4    RELATED WORK

Concurrent with our work, Semeniuta et al. (2018) demonstrated the issues of local n-gram metrics. Their extensive empirical evaluation of GAN models and language models (LM) did not result in evidence of GAN-trained models outperforming on the new and improved global metrics from Cífka et al. (2018). Our analysis further explores this path by examining the performance of these models under a sweep of temperatures. We believe this difference to be of utmost importance, as it is the necessary ingredient towards definitively showing MLE models outperform current GAN variants on quality-diversity global metrics. Work from Ott et al. (2018) thoroughly examines the local beam search strategy for neural machine translation. In their analysis, the authors compare local beam search and generator rejection sampling and find the beam search is quite effective at finding high-likelihood regions. However, their work focuses on conditional text generation, where quality-only metrics measures performance.

Guo et al. (2017) suggest that increasing the temperature at training time leads to more diverse samples. However, we argue that this procedure leads to the opposite outcome as a model can adapt to the temperature change. This would have the net result of lowering the entropy at test time. We discuss this issue and propose a study in Appendix F.

## 5    EMPIRICAL RESULTS

Using our evaluation approach (Section 3), we examine several recent GAN text generation models and compare against an MLE baseline. The experiments consist of two parts: synthetic data generation and long-text generation. We provide strong empirical evidence for both types of data that MLE trained models reliably outperform textual GANs in the quality-diversity space. For these experiments, we only use temperature tuning, as it is the only technique that gives a smooth control over said trade-off whilst being computationally efficient (see Section 5.4).

### 5.1    EXPERIMENTAL DETAILS

The ever-growing body of RL trained language GANs makes writing, running hyperparameter searches and cross-validations on all the variants prohibitively expensive.[2] We implemented our own language GAN, including improvements/functionalities proposed in several text GAN papers. We ensure that each functionality can be toggled during training. Then for each dataset, we ran a hyperparameter search of 300 trials encompassing all possible combinations of said functionalities. We refer to this model as RL-GAN. It is based on SeqGAN (Yu et al., 2017) with additional improvements shown to be useful including: MLE pretraining (Yu et al., 2017), leaky discriminator (Guo et al., 2017), step-level loss instead of sequence level (Fedus et al., 2018), learned baseline to reduce variance (Fedus et al., 2018), regularizing REINFORCE with a maximum-entropy loss (Williams & Peng, 1991) and alternating adversarial loss with MLE loss (Guo et al., 2017).

---

[2]We found the majority of the official implementations to be prone to extreme mode collapse thus making it quite hard to reproduce reported results.

Through comparisons with previously reported results and results obtained via running the official repositories, we show that RL-GAN is the state-of-the-art RL trained GAN. Our MLE model is trained with the same codebase (adversarial training turned off) and only one-sixth of the trials. Moreover, we report SeqGAN and LeakGAN results in the synthetic data experiment. These were obtained with the official repositories that we modified in order to obtain samples at different temperatures. Finally, the SeqGAN, LeakGAN, RankGAN, and MaliGAN results in Figure 4a are taken from (Lu et al., 2018b), which was written by the authors of SeqGAN and LeakGAN.

Finally, we conducted experiments differently for non-RL GANs. The CoT (Lu et al., 2018a) results are obtained with the official repository via a careful hyperparameter search guided by a discussion with the authors. The FM-GAN (Chen et al., 2018) results were obtained with the best performing model that was provided by the authors.

**Selecting temperature sweep range.** We describe how we selected the range of temperatures in the temperate sweeps. We found that in practice, both the quality and the diversity axes induce bounds beyond which results are non-informative. On the diversity dimension, we found no value in *increasing* the entropy of the MLE model because minimizing the forward KL does not lead to an underestimation of the real data entropy. Hence, we decreased the temperatures of the other models until a similar diversity to the MLE model was reached. On the quality dimension, we decreased the temperatures to $\alpha = 0$ but did not report the complete curves for the following reasons. First, with synthetic data (Figure 3) the $NLL_{test}$ explodes when diversity is decreased too much, so we reduce the temperature of the MLE model until the difference in performance compared to the GAN results was evident. Second, in the real data in Figure 4b, we stopped decreasing the temperature when the models achieved a Reverse LM score equal to the perplexity achieved by a 1-gram (unigram) model. This means that the generated dataset is almost non-informative for the (Reverse) language model to learn on i.e., it is as informative as counting tokens. This also coincides with severe mode collapse and a temperature of <0.5 for the MLE model. In Figure 4a, we once more decreased the temperature until the difference in performance of MLE with respect to the other models is unambiguous.

| Model | $NLL_{oracle}$ |
|---|---|
| SeqGAN (Yu et al., 2017) | 8.74 |
| RankGAN (Lin et al., 2017) | 8.25 |
| LeakGAN (Guo et al., 2017) | 7.04 |
| IRL (Shi et al., 2018) | 6.91 |
| MLE ($\alpha = 1.0$) | 9.40 |
| MLE ($\alpha = 0.4$) | 5.50 |
| **MLE ($\alpha = 0.001$)** | **4.58** |

Table 2: $NLL_{oracle}$ measured on the synthetic task *(lower is better)*. All results are taken from their respective papers. An MLE-trained model with reduced temperature easily improves upon these GAN variants, producing the highest quality sample.

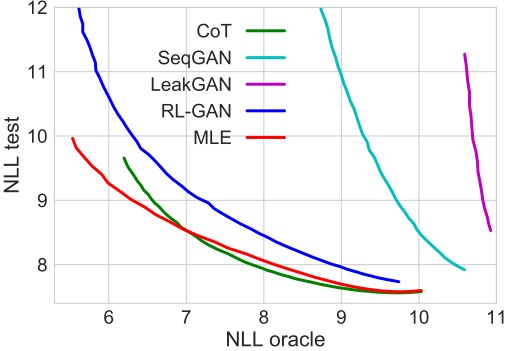

Figure 3: Effect of temperature tuning on the global metrics *(lower is better for both metrics)* for the synthetic task.

## 5.2 SYNTHETIC DATA EXPERIMENT

In the synthetic experiment, we learn a generative model of data produced from a fixed LSTM oracle Yu et al. (2017) with a hidden dimension of 32 with parameters drawn from a standard normal distribution. This allows us to compute a perfect quality metric, the likelihood under the Oracle $NLL_{oracle}$. In Table 2, we see that artificially reducing the temperature at inference achieves state-of-the-art as evaluated by the $NLL_{oracle}$. As we and others have argued, evaluating quality alone is misleading. The MLE-trained model with extremely low temperature will repeatedly output only a single sequence. It is therefore essential to evaluate the resulting sample diversity which we evaluate using the log-likelihood that a generator assigns to held-out data ($NLL_{test}$). We report the result of a temperature sweep in Figure 3.

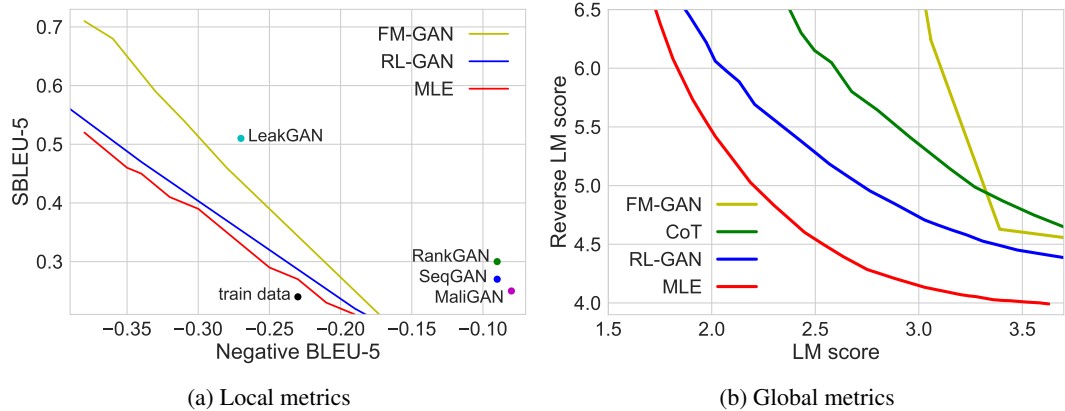

(a) Local metrics                                    (b) Global metrics

Figure 4: Results on the EMNLP 2017 News dataset. *(lower is better for all metrics)*. MLE under a temperature sweep achieves better quality-diversity trade-off compared to the GAN approaches.

Our RL-GAN benchmark is superior to the official SeqGAN and LeakGAN implementations. Nonetheless, MLE outperforms GANs everywhere in the quality-diversity space. Semeniuta et al. (2018) suggest that the best-performing GANs tend to stay close to the solution given by maximum-likelihood training. We find support for this conclusion, as the best performing RL-GAN models have the smallest learning rate and a considerable amount of pretraining. CoT achieves similar performance to MLE, but with dramatically increased algorithmic complexity. This is unsurprising as their objectives are somewhat similar (see Section 4). Alihosseini et al. (2019) replicated this experiment on synthetic data and arrived at the same conclusions.

### 5.3 LONG-TEXT GENERATION

Next, we study long-text generation using EMNLP News 2017. We first compare an MLE model to the reported GAN results on the *local metrics* Negative BLEU and Self-BLEU. Negative BLEU5 and SBLEU-5 are used for Figure 4a and results for (Self-)BLEU-2 to (Self-)BLEU-4 are reported in Appendix D. Again, the conclusions are the same: MLE outperforms all RL GANs considered in the quality-diversity trade-off. Moreover, MLE outperforms FM-GAN, the state-of-the-art non-RL Language GAN. Note that the FM-GAN results were obtained with a combination of temperature tuning and noise reduction in order to achieve the best possible results. We provide additional details in Appendix H. We compare MLE to GANs using recently proposed *global metrics*, the Language Model score (quality) and Reverse Language Model score (diversity+quality) (Cífka et al., 2018; Semeniuta et al., 2018). See Figure 4b for a comparison of the training scheme in quality-diversity space. The conclusion is the same as with BLEU and Self-BLEU: MLE outperforms all GANs everywhere in the quality-diversity space. Further results on this dataset as well as on the IMDB movie reviews (Maas et al., 2011) and WikiText-103 (Merity et al., 2016) datasets are presented in independent follow-up works (d'Autume et al., 2019; Alihosseini et al., 2019) where MLE is also shown to outperform GANs using temperature sweeps.

### 5.4 EMPIRICAL REMARKS

These findings are illustrative: a body of research is predicated on exposure bias as the culprit for poor sample quality. Therefore, under this hypothesis, MLE should be at its weakest on long-text generation tasks. However, our results are evidence that exposure bias is less of an issue than optimization problems arising from GAN training combined with the non-differentiability of the original objective function. There is also another way to interpret these results. It seems that MLE (pre-)training leaves the generator with a better policy according to quality-diversity. However, because GAN training removes entropy from the learned distribution (see Fig. 5 in Appendix **??**), which results in high-quality samples, it can lead one to believe that GAN trained models are superior.

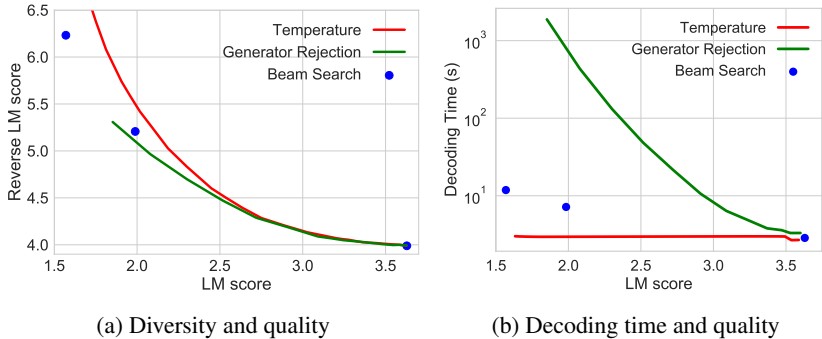

(a) Diversity and quality     (b) Decoding time and quality

Figure 6: Analysis of decoding methods. *(lower is better for all metrics)*. **Left:** Less biased methods provided a better quality/diversity trade-off. **Right:** However, they are computationally much more expensive.

**Methods to approximately reduce the entropy of the joint distribution**   We analyze the different decoding mechanisms as tools to move in the quality-diversity space and as an alternative to temperature sweep. We evaluate these tools by decoding from the best performing MLE model on the EMNLP2017 News dataset. We also consider the different properties of these tools, including their ability to provide smooth control over the quality-diversity trade-off and their computational efficiency. The purpose of this experiment is thus not to find the best performing strategy i.e., a model combined with a decoding method but rather to compare decoding methods. Further, a comparison between MLE and RL-GAN using these approaches is in Appendix I and it supports our prior results.

Figure 6a compares three methods for navigating the quality (Language Model, LM) diversity (Reverse LM) space: temperature sweep, generator rejection,[3] and beam search. We first note that temperature tuning and generator rejection sampling yield a similar trade-off in the high-diversity regime. However, as the bias incurred by temperature tuning grows, so does the gap between both methods. An important finding is that generation rejection sampling gets exponentially slower as the threshold increases (see Figure 6b). It is for this reason that the generator rejection sampling curve doesn't span further in the high-quality part of the space.

Beam search, a less biased method, also seems to offer a better trade-off compared to temperature sweep. However, beam search has a major drawback: unitary increases in beam size leads to drastic leaps in quality-diversity space. In Figure 6a we report beam sizes of 1 (right dot), 2 (middle dot), and 3 (left dot). This study shows that although the less biased methods seem to offer at least small gains in the quality-diversity space, temperature tuning still has important advantages to assess a model's quality-diversity trade-off efficiently. The discreteness of beam search and the computational inefficacy of generator rejection sampling are important drawbacks. However, if one's goal is to obtain the best samples possible, we advocate for their use.

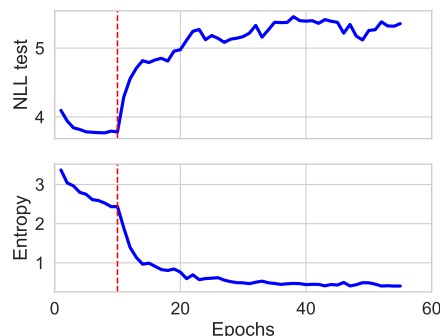

Figure 5: Dotted line indicates the start of GAN training. We notice a clear drop in entropy (spike in $\text{NLL}_{test}$) when moving from maximum-likelihood to adversarial updates.

**GAN Training Reduces Entropy**   Figure 5 shows the evolution of a generator's entropy with respect to epochs. We observe that as soon as adversarial training starts, a drastic entropy reduction occurs. Future work investigates if GAN training changes the learned distribution or simply reduces the entropy in an ineffective way with respect to the quality-diversity trade-off.

---

[3]For computational reasons, generator rejection sampling does not span further in the high-quality regime.

## 6 Discussion

We demonstrate that well-adjusted language models are a strong baseline and that temperature sweeping can provide an unambiguous characterization of model performance in terms of quality and diversity. A well-adjusted language model outperforms the considered GAN variants as evaluated on both local, and more surprisingly, global metrics of quality and diversity. Our temperature sweeping framework shares characteristics with a Receiver Operating Curve. Analogously, if one needed a single scalar to compare NLG models, one could compute the area under the curve and seek the model with the smallest value (lower is better for our considered metrics).

GAN-based generative models have been proven effective on real-valued data. However, there exist many difficult pernicious issues involved in moving to discrete data. These issues must be overcome before they will improve over the strong MLE baselines for unconditional language generation. On the datasets and tasks considered, potential issues caused by exposure bias were less severe than the ones arising from adversarial training combined with learning the non-differentiability of the original objective function. GAN training may prove fruitful eventually, but this research lays forth clear boundaries that it must first surpass.

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

## A  STOCHASTIC BEAM SEARCH.

Beam search is a popular approximate decoding technique. It is computationally expensive but usually leads to higher-likelihood samples than greedy decoding. Here, we focus on its stochastic version, not to be confused with the locally optimal one, Local Beam Search. In Stochastic Beam Search with beam size $k$, words are sampled sequentially starting from the first one. At the first timestep $k$ words are sampled, each represents a hypothesis. At subsequent timesteps, $k$ next words are sampled conditioned on each of the $k$ current hypotheses, resulting in $k^2$ hypotheses. The $k$ most likely are kept and so on. Because an increase in beam size results in higher-likelihood samples, beam size can be leveraged to modulate the quality-diversity trade-off. Moreover, an infinite beam size would result in sampling the most likely sentences under the generator's distribution. This is not what we can expect from setting the temperature to 0, which is equivalent to greedy decoding. For this reason, we can hypothesize that Stochastic Beam Search is less biased regarding the reduction of the joint distribution's entropy. However, unlike temperature tuning, one cannot smoothly trade quality for diversity, as the beam size is a discrete parameter. The more traditional local (deterministic) beam search removes all sample diversity. Thus, it is not useful in our setting.

## B  GENERATOR REJECTION SAMPLING.

*Generator Rejection Sampling* works as follow: generate some sentences; compute their likelihood under the Generator's own distribution; accept/reject the sample given a threshold. The threshold enables the practitioner to modulate the quality-diversity trade-off. Similar to Stochastic Beam Search, increasing the threshold to a maximum level would result in always sampling the most likely sentence. We can again hypothesize that this method achieves a better quality-diversity trade-off compared to temperature tuning, since it is not biased towards having sentences with a lower entropy prefix. However, the running time of this procedure increases significantly as the acceptance threshold lowers, as discussed in Section 5.4. We also experimented with a rejection sampling scheme where the *score* is determined by a discriminator but it did not provide any additional insights.

## C  QUALITATIVE ANALYSIS OF THE SAMPLES

For completeness, we show samples of SeqGAN, LeakGAN, and MLE from both datasets in Table 3. In this experiment, we used the Image COCO dataset (Lin et al., 2014) in order to generate shorter sentences as well. In this case, the samples of all models appear similar. This is because generating captions is a relatively easier task. For the EMNLP2017 News dataset, we would like to point out interesting facts from our samples. In our first sample (1), the model generates the sequence "post Brexit strategy". This is n-gram is not present in the training set. In our second sample (2), the token "leak" and the n-gram "Freedom of Information request" never appear together in the training dataset. The extrapolations show that the model learns a certain level of generalization beyond the training set. Additional samples are presented further in Appendix J.

## D  FULL BLEU AND SELF-BLEU RESULTS

Full BLEU and Self-BLEU results are shown in Table 4.

## E  THE LIMITATIONS OF BLEU

In this section, we want to highlight an important flaw in using BLEU as a proxy for quality. We tuned the temperature in order to find a MLE model with BLEU score equal to the training data's. We show three randomly sampled sentences from the model in Table 6. Although sometimes grammatically correct, the samples lack in semantic and/or global coherence. It seems the generated text has poor information content. Surprisingly, in order to get great samples on a consistent basis, the temperature needs to be reduced to a level where BLEU-5 is twice as large as the training data's. Thus, it seems like BLEU is not always a good proxy of sample quality. Again, we think it is of utmost importance to develop better metrics and modernize NLG's canonical evaluation framework.

| Datasets | Image COCO | EMNLP2017 News |
|---|---|---|
| SeqGAN | (1) A woman is riding a bike on the street next to a bus. | (1) You only certainly might not rush it down for those circumstances where we are when they were the heads, and when she's name. |
| | (2) A silver stove, the refrigerator, sitting in a kitchen. | (2) I think you should really really leave for because we hadn't been busy, where it goes to one," he wrote. |
| LeakGAN | (1) A woman holding an umbrella while standing against the sidewalk. | (1) A man has been arrested at age 28, a resident in Seattle, which was widely reported in 2007. |
| | (2) A bathroom with a toilet and sink and mirror | (2) I also think that's a good place for us, I'm sure that this would be a good opportunity for me to get in touch. |
| MLE | (1) A narrow kitchen with wooden cabinets and white appliances . | (1) The company will be able to provide a post Brexit strategy , which will be published in the coming weeks . |
| | (2) There are several bikes parked in front of a tall building with four cars . | (2) The leak was obtained by a Freedom of Information request , which is based on the number of people claiming to be a victim of fraud . |

Table 3: Samples from the different models on Image COCO and EMNLP2017 WMT News. For SeqGAN and LeakGAN, samples were taken from (Guo et al., 2017). It's the first two samples found in their appendix. For our samples, we reduced the temperature of the model till we achieved similar BLEU scores to the ones reported in (Guo et al., 2017) in order to keep comparison fair.

| | BLEU | | | | Self-BLEU | | | |
|---|---|---|---|---|---|---|---|---|
| | 2 | 3 | 4 | 5 | 2 | 3 | 4 | 5 |
| Training Data | 0.86 | 0.61 | 0.38 | 0.23 | 0.86 | 0.62 | 0.38 | 0.24 |
| SeqGAN Yu et al. (2017) | 0.72 | 0.42 | 0.18 | 0.09 | 0.91 | 0.70 | 0.46 | 0.27 |
| MaliGAN Che et al. (2017) | 0.76 | 0.44 | 0.17 | 0.08 | 0.91 | 0.72 | 0.47 | **0.25** |
| RankGAN Lin et al. (2017) | 0.69 | 0.39 | 0.18 | 0.09 | **0.90** | **0.68** | **0.45** | 0.30 |
| TextGAN Zhang et al. (2017) | 0.21 | 0.17 | 0.15 | 0.13 | 1.00 | 0.98 | 0.97 | 0.96 |
| LeakGAN Guo et al. (2017) | 0.84 | 0.65 | 0.44 | 0.27 | 0.94 | 0.82 | 0.67 | 0.51 |
| MLE ($\alpha = 1.25^{-1}$) | **0.93** | **0.74** | **0.51** | **0.32** | 0.93 | 0.78 | 0.59 | 0.41 |

Table 4: BLEU (left) and Self-BLEU (right) on test data of EMNLPNEWS 2017. (Higher BLEU and lower Self-BLEU is better).

| | BLEU | | | | Self-BLEU | | | |
|---|---|---|---|---|---|---|---|---|
| | 2 | 3 | 4 | 5 | 2 | 3 | 4 | 5 |
| Training Data | 0.74 | 0.53 | 0.34 | 0.22 | 0.90 | 0.75 | 0.58 | 0.42 |
| SeqGAN Yu et al. (2017) | 0.75 | 0.50 | 0.29 | 0.18 | 0.95 | 0.84 | 0.67 | 0.49 |
| MaliGAN Che et al. (2017) | 0.67 | 0.43 | 0.26 | 0.16 | 0.92 | 0.78 | 0.61 | 0.44 |
| RankGAN Lin et al. (2017) | 0.74 | 0.47 | 0.26 | 0.16 | 0.96 | 0.88 | 0.76 | 0.62 |
| TextGAN Zhang et al. (2017) | 0.59 | 0.46 | 0.28 | 0.21 | 0.94 | 0.93 | 0.80 | 0.75 |
| LeakGAN Guo et al. (2017) | **0.74** | **0.52** | **0.33** | **0.21** | 0.93 | 0.82 | 0.66 | 0.51 |
| MLE | **0.74** | **0.52** | **0.33** | **0.21** | **0.89** | **0.72** | **0.54** | **0.38** |

Table 5: BLEU (left) and Self-BLEU (right) on test data of Image COCO. (Higher BLEU and lower Self-BLEU is better).

## F  Issues of Varying Temperature During Training

As a penultimate experiment, we analyze the effect of changing the temperature at *training* instead of inference. Guo et al. (2017) suggested that increasing the temperature at training time leads to more diverse samples. However, we argue that this procedure leads to the opposite outcome as a model can adapt to the temperature change. This would have the net result of lowering the entropy at test time. To examine this, we trained 30 GANs maintaining everything constant except training temperature. Negative BLEU-5 against SBLEU-5 are plotted in Figure 7. The darker the dot, the higher the $\alpha$ and consequently the temperature. As we hypothesize, models trained with increased

| MLE | | (1) He explained that the Government ' s plan to cut tax on unemployment was 3 . 3 percent lower than forecast for the first increase of 16 percent in 2015 , the fastest rate in the state since 2004 . |
| $\alpha$ $1.05^{-1}$ | $=$ | (2) On the policy , it ' s no more than the amount of money we have of the decades and Senate of our assets . |
| | | (3) They say it was possible supporting the Scottish government to make the changes as secret free environment based on competition . |

Table 6: Three randomly sampled sentences from our model with closest BLEU scores to the training set's. The sentences have poor semantics or global coherence. They are also not perfect grammatically speaking.

temperature at training time adapted to the change and the net result was a colder temperature at inference (hence reduced diversity). We therefore recommend only adjusting the temperature at inference. One should consider other techniques to facilitate exploration during training.

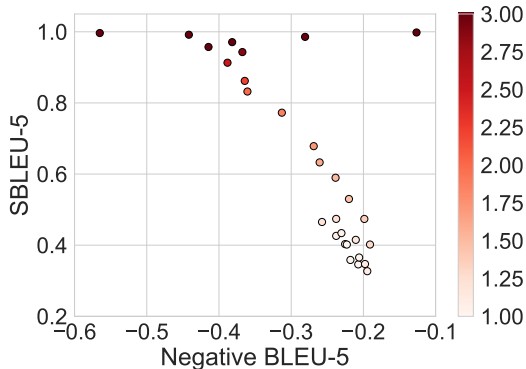

Figure 7: Negative BLEU-5 on test data against SBLEU5 for models with different temperature applied at training time. The redder the dot, the higher the $\alpha$ i.e. more pressure to increase entropy. From these results, it is evident that one should not increase the temperature at training time as the models adapts and the net results is mode collapse.

## G  DISCRIMINATOR REJECTION SAMPLING

We experimented with a rejection sampling technique base on the Discriminator's signal. For this method, we train a Discriminator on the Generator's samples without ever updating the Generator. Next, we accept/reject samples in a similar fashion to Generator Rejection Sampling however now the score is determined by the Discriminator. The higher the threshold, the more confident the Discriminator needs to be about the realness of the data. This should increase the quality of the samples with the usual downside of reducing diversity. We show results with a Discriminator having the same architecture as the Generator (Discriminator Rejection) and the best Discriminator found with an hyper parameter search of 100 trials (Best Discriminator Rejection. Results are shown in Figure 8. Note that the domain of the figure is much smaller that in Figure 4b. The reason is that this approach can't move the models further in quality-diversity space. For this reason, we do not advocate for the use of this approach as a means to obtain high quality samples.

## H  MODULATING QUALITY/DIVERSITY FOR FM-GAN

For FM-GAN, we observed that lowering the temperature to 0 didn't completely remove all the stochasticity in the generations. This is because FM-GAN samples noise prior to sampling the first word. This is similar to how image generation GANs are trained. Precisely, $z \sim N(0, 1)$ is sampled once and concatenated to every token of a generated sentence. Because temperature tuning wasn't covering enough of the quality/diversity space, we also reduced the variance of the noise as quality modulating tool. Results are show in Figure 9.

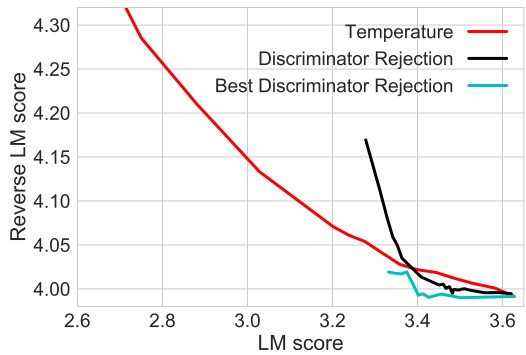

Figure 8: Discriminator Rejection Sampling is not a great tool to navigate in quality-diversity space.

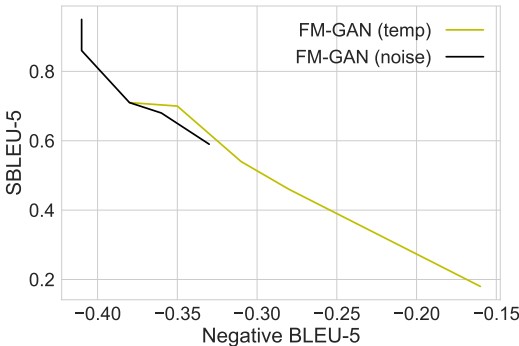

Figure 9: Different strategies to modulate the quality/diversity trade-off in FM-GAN.

# I  RL-GAN VERSUS MLE WITH OTHER DECODING MECHANISMS

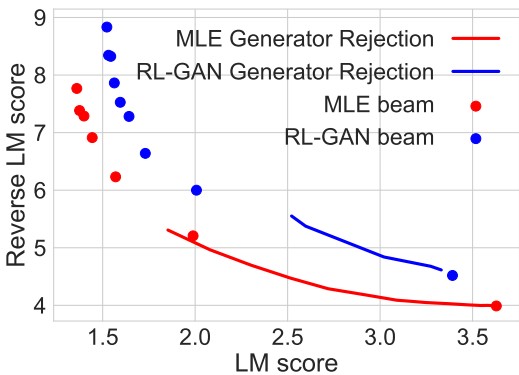

Figure 10: MLE still outperforms RL-GAN under different decoding mechanisms.

## J  EMNLP 2017 NEWS SAMPLES

We present additional samples for EMNLP 2017 News dataset.

|  | EMNLP2017 News |
|---|---|
| SeqGAN | You only certainly might not rush it down for those circumstances where we are when they were the heads , and when she s name . |
| | I think you should really really leave for because we hadn t been busy , where it goes to one , he wrote . |
| | All the study knew was that they are , so they continue to provide support service and it doesn t exist . |
| | It can say become up with nothing sales have reached the charge for the other any evidence that been virtually well below the $ 800 . |
| | Three times before the start of the season is much early on 2015 we are in the third training every year . |
| | That s the idea of strength that decision they said, we haven t already lost four or seven, or Liverpool s team . |
| | That is not the time for the cost of changing the system and it was pushing for $ 20 million . |
| | We had to take it a good day for a military , but nearly 6 , 000 ] and prepare for them through . |
| | I actually didn t tell the background check the difference after my hour was to be recalled . . . and it was great . |
| | We are thinking about 40 , 000 and jobs in what is wrong in the coming and you know . |
| | That is out how working you can t set out some pretty tight . . . or what I m going through . |
| | I wanted to be made you decided to have a crisis that way up and get some sort of weapon , not much to give birth to for an American room . |
| | She had been fined almost 200, 000 with couple of asylum seekers in Syria and Iraq . |
| | Perhaps not , in looking for , housing officials would help the frustration of Government , with an FBI shortly before 2020 . |
| | Once we got to real show for the young man since I m sure she went to love it just , whether to be late later last year . |
| | But , after a holiday period we might have to go on a total - out debate like that could have happened to us . |

Table 7: Samples from SeqGAN taken from Guo et al. (2017).

|          | EMNLP2017 News |
|----------|----------------|
| LeakGAN | A man has been arrested at age 28 , a resident in Seattle , which was widely reported in 2007 . |
|          | I also think that s a good place for us , I m sure that this would be a good opportunity for me to get in touch . |
|          | What is the biggest problem for Clinton is that Donald Trump will be in the race and he s unlikely to be the nominee . |
|          | We re going to do and we re going to put it out and get the ball , he said . |
|          | I would be afraid to blame the girls to go back but I was just disappointed with the race , he said. |
|          | I m not going to work together with a different role and we can win the game , he added . |
|          | The couple s lives are still missing and they have been killed in the city s way to play against them , and because I came out there . |
|          | For the last three years , we ve got a lot of things that we need to do with this is based on the financial markets . |
|          | Don t ask me , but I know , if I ll be able to be out of Hillary Clinton , I think it s being made for the Congress. |
|          | I am proud to be able to move forward because we don t have to look at about , he said . |
|          | That s why we re the most important people for the African American community and we ve made a good response . |
|          | But the move will be only in a fight against them, as well as likely to prevent an agreement to remain in the EU . |
|          | The American Medical Association said that the militants had been arrested in connection with the murder of the same incident. |
|          | The two - year - old girl has been charged with a suspect who was in the vehicle to the police station. |
|          | It is hard to buy on the Olympics , but we probably don t see a lot of it. |
|          | I m not going to be very proud of the other countries , he said . |
|          | He said the U . N . intelligence industry will not comment on the ground , which would be sensitive to the European Union . |
|          | I take my work in the days , but I would have to go down on Wednesday night . |

Table 8: Samples from LeakGAN taken from Guo et al. (2017).

|  | EMNLP2017 News |
|---|---|
| MLE | The UN Security Council is a major concern for the U . S . government , as well as a NATO ally in the Syrian civil war . |
|  | A spokesman for the Met Office said the death toll was only slightly higher than the previous year , according to the report . |
|  | But I hope that at the end of the day , I ' m going to give her the best chance to go to the gym and go out and play . |
|  | The man , who cannot be named , said that he had never had sex with him , and he didn ' t want to see him . |
|  | And it ' s just one of those things that I have to say , I ' m a Democrat , and I ' m a conservative . |
|  | The bank is now the fastest growing market in the world and it is a significant change in the economy . |
|  | The two men , aged 20 and 22 , were arrested and charged with the murder of a man , a police officer . |
|  | The company will be able to provide a post Brexit strategy , which will be published in the coming weeks . |
|  | She said she had been on the wrong side of the road and was finally caught in a car accident and was taken to hospital . |
|  | I don ' t think he ' s even a good player , he said , but he ' s got a good chance to win the game . |
|  | I don ' t know what the future holds but I ' m sure it will be a good thing . |
|  | It ' s a very important step forward , and we ' re going to be able to get the right results . |
|  | The driver of the vehicle , who was inside the vehicle , was taken to hospital for treatment , but said he was not aware of the incident . |
|  | The leak was obtained by a Freedom of Information request , which is based on the number of people claiming to be a victim of fraud . |
|  | The former secretary of state has made a major speech in New York , where she ' s running for president . |
|  | The US economy grew at a record low of 1 . 6 percent in 2014 , and the unemployment rate has fallen by 0 . 9 percent . |
|  | The new rules are put into the hands of a member of the police , and the public is not aware of the situation . |
|  | The World Health Organization said a number of people were killed in the attack , according to the Pentagon . |
|  | The study also found that women who are not particularly vulnerable to women ' s health problems are more likely to commit suicide . |

Table 9: Samples from our MLE with temperature to match BLEU scores reported in Guo et al. (2017)

