# OpenReview forum: "Language GANs Falling Short"
_ICLR.cc/2020/Conference — Accept (Poster)_

### Official Review · AnonReviewer1 · 2019-10-24
**Official Blind Review #1**

**Rating:** 8

**Review:**

Recently many language GAN papers have been published to overcome the so called exposure bias, and demonstrated improvements  in natural language generation in terms of sample quality, some works propose to assess the generation in terms of diversity, however, quality and diversity are two conflicting measures that are hard to meet. This paper is a groundbreaking work that proposes receiver operating curve or Pareto optimality for quality and diversity measures, and shows that simple temperature sweeping in MLE generates the best quality-diversity curves than all language GAN models through comprehensive experiments. It points out a good target that language GANs should aims at.

For the experiments on long-text generation using EMNLP news 2017, it is not clear how the data is partitioned as training data, validation data and test data to get the results in Figures 4(a), moreover for the results for LeakGAN, RankGAN SeqGAN, MaliGAN, it seems that they are copied from other papers, but again how the data set is partitioned is not clear  for example in SeqGAN's paper, and most likely, the data is partitioned in a different way, so the results are not comparable. The authors should run the code and get the results on it own.

An important RL-free language GAN paper is missing,
Zhongliang Li, Tian Xia, Xingyu Lou, Kaihe Xu, Shaojun Wang, Jing Xiao: Adversarial Discrete Sequence Generation without Explicit NeuralNetworks as Discriminators. AISTATS 2019: 3089-3098.
This paper directly and adversarially train a language model without MLE pre-training and obtains good results, it is better to compare the results.

Typo: page 1, the second line from bottom, as a computationally, not an

**Experience Assessment:**

I have published one or two papers in this area.

**Review Assessment: Checking Correctness Of Derivations And Theory:**

I carefully checked the derivations and theory.

**Review Assessment: Checking Correctness Of Experiments:**

I carefully checked the experiments.

**Review Assessment: Thoroughness In Paper Reading:**

I read the paper thoroughly.

---

> ### Author Response · Authors · 2019-11-07
> **Response to Reviewer #1**
>
> Thanks for your encouraging comments!
>
> Thanks for the suggestion to clarify the data partition. Some results are taken from another paper, but we made sure the data partition was the same. The text generation benchmarking platform Texygen https://github.com/geek-ai/Texygen was used to generate the performance results we copied from https://arxiv.org/pdf/1803.07133.pdf . This platform and this paper were developed and written by the same lab who proposed SeqGAN and LeakGAN. We used the dataset generating and metrics computing scripts from Texygen to ensure comparability. Thus, the reported results in Figure 4(a) are legitimate.
>
> We thank the reviewer for pointing out the missing citation, which we will add to the next version of the paper. It is however not possible at the time to provide a fair comparison to its results, as the paper only reports quality metrics (NLL_oracle and BLEU). Furthermore, we could not find a publically available implementation. We would be willing to compare with this methodology if either condition was addressed.
>
> Thanks for pointing out the typo.

---

> > ### Comment · AnonReviewer1 · 2019-11-08
> > **Official Blind Review #1**
> >
> > My experience tells me that SeqGAN and LeakGAN are not as bad as shown in Figure 4(a), that's why I suspect   the data partition is not the same. The lab published a series of GANs paper over several years, so they might change the data partition. Please contact this lab to make sure the data partition is the same, if not, please re-do the experiments.
> >
> > Regarding to the results of Li et al's work, a paper titled "A quality-diversity controllable GAN for text generation" (https://openreview.net/forum?id=rJlTXxSFPr, submitted this conference) gives some results of Li et al's and does report diversity metrics, but I'm not sure the data partition is the same as yours.

---

> > > ### Author Response · Authors · 2019-11-11
> > > **Response to Reviewer #1**
> > >
> > > Thanks again for helping us with our empirical results. The original EMNLP2017 News dataset is much bigger than the one in Texygen used by the SeqGAN/LeakGAN lab and by us. This could be the reason why the reviewer as seen better results on this dataset. We found further evidence in “CoT: Cooperative Training for Generative Modeling of Discrete Data” (v1: https://arxiv.org/pdf/1804.03782v1.pdf), another paper authored by the same lab, which indicates that our results are in line with the literature. Specifically, they report a BLEU5 and SBLEU5* of 0.10 and 0.27 for SeqGAN and 0.27 and 0.53 for LeakGAN. This is close to our 0.09 and 0.27 for SeqGAN and 0.27 and 0.51 for LeakGAN.
> > >
> > > Furthermore, preprocessing can have an impact on the results. E.g. in “A Quality-Diversity Controllable GAN for Text Generation”, the dataset is processed differently and their results are different than ours. Nonetheless, they arrived at the same conclusion regarding MLE outperformance over GANs.
> > >
> > > We, however, found that Texygen and the other papers from the same lab don’t specify a validation set. Thus, the reviewer might be right that some noise could have been inserted in our empirical results. We have opened an issue on the Texygen repository https://github.com/geek-ai/Texygen/issues/42 . We emailed the authors as well. We will re-do the experiments if there is a discrepancy is the data partition.
> > >
> > > Nevertheless, the comparison between MLE and our RL-GAN (which includes SeqGAN, LeakGAN and other tricks from other Language GANs) was performed with a single codebase (ours) and data partition. Thus, our conclusion that MLE training is superior to GAN training when considering the full quality-diversity tradeoff holds for our data partition.
> > >
> > >
> > > * we multiplied the values in Table 3 of v1 https://arxiv.org/pdf/1804.03782v1.pdf by 0.24, which is the SBLEU5 of the real data.

---

> > > > ### Comment · AnonReviewer1 · 2019-11-15
> > > > **Official Blind Review #1**
> > > >
> > > > Thanks for the clarification.
> > > >
> > > > Regarding to the results in "A Quality-Diversity Controllable GAN for Text Generation", Li et al's correspond to those when pi = 0,5, and it seems that they are comparable to yours for the synthesis data,  but not for the real data. Moreover MLE outperformance over GANs is not true for the synthesis data and Coco dataset when considering pi = 0.5.

---

> > > > > ### Author Response · Authors · 2019-11-15
> > > > > **Response to Reviewer #1**
> > > > >
> > > > > thanks for pointing this out, we'll carefully study this other submission and will consider citing it

---

### Official Review · AnonReviewer2 · 2019-10-26
**Official Blind Review #2**

**Rating:** 6

**Review:**

This paper concerns the limitation of the quality-only evaluation metric for text generation models. Instead, a desirable evaluation metric should not only measure the sample quality, but also the sample diversity, to prevent the mode collapse problem in gan-based models generation. The author presents an interesting, but not too surprising finding that, tuning the temperature beam search sampling consistently outperform all other GAN/RL-based training method for text generation models. The idea of sweeping temperature during beam search decoding is not new in the NLP community, which limits the novelty of this paper. What’s more, some parts of the experiment results is also somehow not new, in the sense that the SBLEU vs Negative BLEU tradeoff curve is also shown in [1,2,3,4].

[1] Jointly measuring diversity and quality in text generation models, 2019
[2] Training language gans from scratch, 2019
[3] On accurate evaluation of gans for language generation, 2018
[4] Towards Text Generation with Adversarially Learned Neural Outlines, 2018

I would love to increase my score if the author could address the following comments:
(1) Are the comparing methods, say MLE models and other GAN-based models, have the similar number of model parameters? It is not clear from the paper. Otherwise, one can use a 12/24 layer Transformer-XL to have dominative performance?
(2) Since this is an empirical study paper. It would be great if this paper can also present more SoTA models trained by MLE such as Transformer-XL on more challenging datasets, such as Wikitext-2 or Wikitext-103. In this kind of large vocabulary datasets, I think the RL/GAN-based training methods would easily breakdown, and far worse than MLE-based training.
(3) To make the empirical study more comprehensive, the author could perhaps evaluate with the n-gram and FED metric.


**Experience Assessment:**

I have read many papers in this area.

**Review Assessment: Checking Correctness Of Derivations And Theory:**

I assessed the sensibility of the derivations and theory.

**Review Assessment: Checking Correctness Of Experiments:**

I assessed the sensibility of the experiments.

**Review Assessment: Thoroughness In Paper Reading:**

I read the paper at least twice and used my best judgement in assessing the paper.

---

> ### Author Response · Authors · 2019-11-07
> **Response to Reviewer #2**
>
> Thanks for your time reviewing and help improving our paper!  Thank you for pointing out this related work. Indeed, [1,2] are related, however, we note that our work preceded them. While this fact is easily verifiable, we do not provide the outside evidence here to preserve anonymity.  Our temperature evaluation framework has been successfully employed by the community as discussed in our submission.  We additionally encourage the reviewer to revisit Page 4, Paragraph 2 reproduced for convenience here:
>
> “In the next section we show, using temperature sweep, that MLE models consistently outperform the new proposed GAN variants everywhere in the quality-diversity space. MLE performs equally on synthetic data to CoT and outperforms it on real data, whilst being computationally and algorithmically less complicated. Our results are further validated in independent follow-up works (d’Autume et al., 2019 [2]; Alihosseini et al., 2019 [1]) where temperature sweeps show MLE outperforming GANs.”
>
> Regarding the other two papers, [3] was concurrent work that analyzes the BLEU and SLEU tradeoff but does not show any temperature curves. They show something akin to our Figure 2, i.e. a single point for each model in quality-diversity space. It’s important to realize, as explained in our introduction (see Figure 1) that this evaluation can lead to inconclusive results. This is one of the contributions of our work. Note that [4] similarly does not propose temperature sweeps. They do report results at two different temperatures (0.5 and 1.0). However, their conclusion is that a pre-trained general-purpose sentence encoder can be leveraged in unconditional text generation. Our conclusion is that MLE training is still superior to GAN training when considering the full quality-diversity tradeoff.  Though this may seem like a simple point, it is an important message for the community that may impact many future works.
>
> Comment (1): Thanks for the suggestion to clarify the model sizes. We will add these details to the paper!  For both MLE and RL-GAN, we hyperparameter searched for the best depth {1,2} and width {128, 256, 512} of the LSTM. More details about hyperparameter values can be found in the hyperparameter search script https://github.com/AnonSubmitter2/iclr2020/blob/master/cc_folder/news_rs.py. So the comparison is equivalent on a hyperparameter computational budget.  Furthermore, the text GANs are O(2x) as large due to the discriminator and generally require longer training.
>
> Comment (2): We definitely agree RL/GAN-based methods could easily breakdown on more challenging datasets and harder to optimize architectures like Transformer-XL. As you correctly point out, policy-gradient and reinforcement learning methods in enormous action spaces are problematic.  We therefore agree and also expect given all the evidence we have presented, that the MLE-based training will be superior to GAN-based training - inline with the central message of our paper.  Our goal here is not to construct a new Transformer-based text GAN, but rather, compare all the SOTA GAN methods proposed and published.
>
> Comment (3): Could you please clarify your request regarding n-gram metrics?  BLEU and SBLEU are already n-gram metrics.  In the main part of the paper, BLEU-5 and SBLEU-5 are reported, which are weighted averages of n-gram overlaps (n from 2 to 5). BLEU2-4 and SBLEU2-4 are shown in the Appendix. If by FED the reviewer meant Frechet InferSent Distance (FID), it was shown in [3] that Reverse LM score is a better metric (see their Figure 2). Although both metrics are sensitive to mode collapse and word dropping, FID is insensitive to word swapping whereas Reverse LM score is not (see page 7 paragraph 2 of their paper).

---

> > ### Author Response · Authors · 2019-11-14
> > **Response to Reviewer #2**
> >
> > As the rebuttal period draws to a close we wanted to ensure we have resolved all your concerns and answered any remaining questions.  Regarding your comments on GAN + Transformer training, we agree that is an interesting open-research direction, but this is not the goal of our research.  In this work, we establish strong baselines for all text GANs but do not seek to design new text GAN variants.  To our knowledge, there are no successful applications of GAN + Transformers and this is an area of open research.  However, if and when a GAN + Transformer variant is released, we will eagerly benchmark it using our temperature sweep methodology, as we did with other released text GAN variants.
> >
> > Do you have any further questions? Thanks again for your review!

---

> > ### Comment · AnonReviewer2 · 2019-11-15
> > **Thanks for clarification**
> >
> > Thanks for your response. My comments (1) and (3) are resolved.
> >
> > For comments (2), I am not expecting to see a new Transformer-based text GAN. Instead, I am suggesting to conduct an experiment comparing the same generative model (i.e. LSTM with the same number of parameters) trained by MLE and GAN loss, on more challenging language modeling datasets, such as PTB, Wikitext-2, or Wikitext-103.

---

> > > ### Author Response · Authors · 2019-11-15
> > > **More Challenging Datasets**
> > >
> > > Great - glad we could clarify comments 1 and 3!
> > >
> > > Ah! Yes, we misunderstood your suggestion.  We can certainly run on a more challenging language modeling before the camera ready deadline (but not before end of rebuttal tomorrow).   Our expectation, given the mode collapse documented in Fedus et al. (2017) on PTB and IMDB movies dataset (see Figure 2 in Appendix C.4), is that our conclusion via temperature sweeping will hold.  Will that resolve your second comment?

---

> > > > ### Comment · AnonReviewer2 · 2019-11-15
> > > > **Official Blind Review #2**
> > > >
> > > > Yes, that will resolve my concern. I increased the score from 3 to 6.

---

> > > > > ### Author Response · Authors · 2020-02-08
> > > > > **results on more challenging datasets**
> > > > >
> > > > > We found that [1] and [2] were already performing temperature sweeps on harder datasets, namely IMDB and Wikitext-103 respectively. Furthermore, [1] used the Texygen platform, (owned by SeqGAN and LeakGAN group) to perform the experiments. Both papers arrived at the same conclusions as us, as previously discussed.
> > > > >
> > > > > Thus, we kindly ask the reviewer if we could be relieved of having to re-run these experiments. Has the results are already available to the community, it would seem like a waste of resources.
> > > > >
> > > > > Thanks

---

### Decision · Program_Chairs · 2019-12-19

**Decision:**

Accept (Poster)

**Comment:**

Main content:

Blind review #1 summarizes it well:

Recently many language GAN papers have been published to overcome the so called exposure bias, and demonstrated improvements  in natural language generation in terms of sample quality, some works propose to assess the generation in terms of diversity, however, quality and diversity are two conflicting measures that are hard to meet. This paper is a groundbreaking work that proposes receiver operating curve or Pareto optimality for quality and diversity measures, and shows that simple temperature sweeping in MLE generates the best quality-diversity curves than all language GAN models through comprehensive experiments. It points out a good target that language GANs should aims at.

--

Discussion:

The main reservation was the originality of the idea of using temperature sweep in the softmax. However, it turns out this idea came from the authors in the first place, which they have not been able to state directly due to the anonymity requirement. Per the program chair's instruction to direct this to the area chair, I think this has been handled correctly.

--

Recommendation and justification:

This paper should be accepted. It provides readers with insight in that it illuminates a misconception of how important exposure bias has been assumed to be, and provides a less expensive MLE based way to train than GAN counterparts.